# Cancer Stem Cells and Somatic Stem Cells as Potential New Drug Targets, Prognosis Markers, and Therapy Efficacy Predictors in Breast Cancer Treatment

**DOI:** 10.3390/biomedicines9091223

**Published:** 2021-09-14

**Authors:** Olga Pershina, Natalia Ermakova, Angelina Pakhomova, Darius Widera, Edgar Pan, Mariia Zhukova, Elena Slonimskaya, Sergey G. Morozov, Aslan Kubatiev, Alexander Dygai, Evgenii G. Skurikhin

**Affiliations:** 1Laboratory of Regenerative Pharmacology, Goldberg ED Research Institute of Pharmacology and Regenerative Medicine, Tomsk National Research Medical Centre of the Russian Academy of Sciences, Lenin, 3, 634028 Tomsk, Russia; ovpershina@gmail.com (O.P.); Nejela@mail.ru (N.E.); angelinapakhomova2011@gmail.com (A.P.); artifexpan@gmail.com (E.P.); mashazyk@gmail.com (M.Z.); amdygay@gmail.com (A.D.); 2Stem Cell Biology and Regenerative Medicine Group, School of Pharmacy, University of Reading, Whiteknights Campus, Reading RG6 6AP, UK; d.widera@reading.ac.uk; 3Department of General Oncology, Cancer Research Institute Tomsk NRMC, 5 Kooperativny Street, 634009 Tomsk, Russia; slonimskaya@rambler.ru; 4Institute of General Pathology and Pathophysiology, 125315 Moscow, Russia; biopharm@list.ru (S.G.M.); niiopp@mail.ru (A.K.)

**Keywords:** breast cancer, ALDH1, cancer stem cells, somatic stem cells, biomarkers

## Abstract

New drug targets, markers of disease prognosis, and more efficient treatment options are an unmet clinical need in breast cancer (BC). We have conducted a pilot study including patients with luminal B stage breast cancer IIA–IIIB. The presence and frequency of various populations of cancer stem cells (CSC) and somatic stem cells were assessed in the blood, breast tumor tissue, and normal breast tissue. Our results suggest that patients with BC can be divided into two distinct groups based on the frequency of aldehyde dehydrogenase positive cells (ALDH1^+^ cells) in the blood (ALDH1^hi^ and ALDH1^low^). In the ALDH1^hi^ cells group, the tumor is dominated by epithelial tumor cells CD44^+^CD24^low^, CD326^+^CD44^+^CD24^−^, and CD326^−^CD49f^+^, while in the ALDH1^low^ cells group, CSCs of mesenchymal origin and epithelial tumor cells (CD227^+^CD44^+^CD24^−^ and CD44^+^CD24^−^CD49f^+^) are predominant. In vitro CSCs of the ALDH1^low^ cells group expressing CD326 showed high resistance to cytostatics, CD227^+^ CSCs of the ALDH1^hi^ cells group are sensitive to cytostatics. Epithelial precursors of a healthy mammary gland were revealed in normal breast tissue of patients with BC from both groups. The cells were associated with a positive effect of chemotherapy and remission in BC patients. Thus, dynamic control of their presence in blood and assessment of the sensitivity of CSCs to cytostatics in vitro can improve the effectiveness of chemotherapy in BC.

## 1. Introduction

Despite advances in diagnosis and treatment, breast cancer (BC) remains the leading cause of cancer death in women [1]. More than 1 million new cases of the disease are registered annually [2]. After the diagnosis of BC, prognosis of complications and the choice of optimal drug therapy are crucial [3]. Traditional prognostic measures are the definition of metastases in the lymph nodes, the size of the tumor, and the type of differentiation of tumor cells [1,3]. When searching for tumor markers, much attention is paid to the tumor subtype, which is determined by the presence (or absence) of the estrogen receptor (ER), progesterone receptor (PR), and the protein associated with the human epidermal growth factor receptor (HER2). In addition, the spectrum of prognostic markers includes Ki-67, urokinase plasminogen activator, as well as plasminogen activator inhibitors (PAI-1) [4]. Immunohistochemical assessment of these antigens was used as the basis of diagnostic tests to determine the likelihood of disease recurrence (MammaPrint, Prosigna, Oncotype DX, EndoPredict) [4]. However, the definition of these parameters is insufficient for personalized treatment of patients with BC. In this regard, the search for effective prognosis biomarkers and predictors of breast cancer therapy is relevant. For the diagnosis and dynamic control of BC in the blood of patients, circulating tumor cells (CTC) are determined [5]. However, breast cancer progression and metastasis are associated with the presence of CSCs [5]. At the same time, breast tumor CSCs are an extremely heterogeneous population [5]. Thus, we have hypothesized that CSCs can be used as an early diagnostic marker and a basis of personalized treatment of BC.

According to modern concepts, inflammation is central to all types of cancer. It is estimated that up to 20% of human cancers are initiated by the inflammatory process [6]. Inflammation initiates the tumor and promotes tumor growth thus playing a central role at all stages of tumorigenesis, including progression, invasion, angiogenesis, and metastasis [7,8,9,10].

In addition to inflammatory cells and proinflammatory cytokines, hematopoietic stem and progenitor cells play an important role in the development of inflammation [11,12,13]. Hematopoietic stem cells (HSCs) can be present in the tumor stroma and differentiate into stromal cells. Moreover, cells derived from hematopoietic stem cells play an important role in the development of tumors [14]. However, this aspect is practically not considered in clinical practice. HSCs act as a potential marker of inflammation and as a target for anti-inflammatory therapy.

Other important players in tumor progression are endothelial progenitor cells [15]. Angiogenesis is observed at the preinvasive stage of the tumor process. This is indicated by the formation of microvessels around the ducts, and the ducts are filled with proliferating epithelial cells [16]. The process of neovascularization is activated simultaneously with the growth of the tumor. New vessels not only help to cope with the growing metabolic needs of the tumor, but also contribute to the spread of the tumor and metastasis. An unfavorable prognosis of breast cancer correlates with increased microvascular density and a high level of pro-angiogenic factors [17].

In this pilot study, we studied CSCs in breast cancer patients in order to find new diagnostic biomarkers and predictors of complications. In addition, we have studied somatic stem cells (SSC), hematopoietic stem cells, and CD309^+^ endothelial cells.

## 2. Materials and Methods

### 2.1. Patients

The study included 12 patients with IIA–IIIB (T1–4N0–3M0) breast cancer of the luminal B and triple-negative molecular subtypes aged from 38 to 66 years (average age of 50.6 ± 3.02 years) who received treatment at the Cancer Research Institute of Tomsk NRMC (Tomsk, Russia) from 2017–2018. Imaging of the primary breast lesion was performed by mammography and ultrasonography. An immunohistochemical study was conducted to determine the molecular subtype of the tumor before treatment. The luminal B subtype of breast cancer was defined as ER+, PR+ or −, Ki67 > 30%, and all patients with the luminal B subtype were HER2-negative. Some patients showed no expression of ER, PR, and HER2, and they were classified as a triple-negative subtype. Histological diagnosis was confirmed for all samples. Appendix A presents the clinical indicators of tumors in all 12 patients. Blood samples from 10 healthy women of similar age were used as control.

This was a pilot investigation. Informed consent was obtained from all individual participants included in the study. All procedures performed in the studies involving human participants were in accordance with the ethical standards of the institutional and/or national research committee and with the 1964 Helsinki declaration and its later amendments or comparable ethical standards.

### 2.2. Design of Investigation

Blood samples were obtained from patients one day before surgery (Figure 1). Breast tumor and normal breast tissue were obtained from patients on the day of surgery.

### 2.3. Isolation of Blood Mononuclear Cells

Lympholyte-H (CEDARLANE, Netherlands, Cedarlane Laboratories, Cat#CL5015) protocol was used for the elimination of erythrocytes and dead cells from human blood and receiving mononuclear cells.

### 2.4. Dissociation of Breast Cancer Cells and Mammary Tissue Cells

Human breast tumor and mammary tissue samples were dissociated overnight with collagenase/hyaluronidase enzymes (StemCell Technologies, Vancouver, Canada, BC, Cat#079128). The human mammary organoids, obtained through differential centrifugation, were then dissociated into a single cell suspension using trypsin/EDTA (StemCell Technologies, Vancouver, Canada, Cat#07901), dispase (StemCell Technologies, Vancouver, Canada, Cat#07913) and DNase (StemCell Technologies, Vancouver, Canada, Cat#07900) enzymes.

### 2.5. Flow Cytometry

Mononuclear cells from blood, human mammary tissue, and breast cancer tissue were isolated as described above and the expression of surface markers on mononuclear cells was analyzed using flow cytometry. Fc-receptors were blocked by pre-incubation of the cells with unconjugated anti-CD16/CD32 antibodies for 10 min (eBioscience, San Diego, CA, USA, Cat# 464219, 1/100 dilution) in 50 μL of 0.1% saponin (Sigma-Aldrich, St. Louis, MO, USA, Cat# S4521) and 1% BSA (Sigma-Aldrich, St. Louis, MO, USA, Cat# A3059-100G) in phosphate buffered saline (PBS) per tube. After the pre-incubation, cells suspensions were stained with fluorophore-conjugated monoclonal antibodies.

#### 2.5.1. Measurement of ALDH Activity

An aldehyde dehydrogenase-based cell detection kit (StemCell Technologies, Vancouver, BC, Canada, Cat#01700) was used to determine ALDH1 enzymatic activity in the blood, breast tumor, and normal breast tissue. Cells were suspended in aldefluor assay buffer and incubated with the ALDH enzyme substrate, BODIPY-aminoacetaldehyde (BAAA), for 40 min at 37 °C. As a control, cells were also treated with diethylaminobenzaldehyde (DEAB), an inhibitor of ALDH enzyme activity. Fluorescence was determined using a BD FACS Canto II flow cytometer and analyzed using FACSDiva software (BD Biosciences).

#### 2.5.2. Detection of Hemopoietic Stem Cells

A two-color research reagent CD45/CD34 (Becton Dickinson, San Jose, CA, USA, Cat# 341071) was used to determine the presence of HSCs. The reagent contains FITC-labeled CD45, clone 2D1, and PE-labeled CD34, clone 8G12. We determined HSC according to the ISHAGE protocol [18].

#### 2.5.3. Detection of Cancer Stem Cells and Somatic Stem Cells

Cell suspensions were stained with fluorophore-conjugated monoclonal antibodies. The following antibodies were used for cell surface staining of mononuclear cells derived from blood and tumors of patients: PE-Cy7-conjugated anti-CD24, APC-conjugated anti-CD44, APC-H7-conjugated anti-CD45, PE-conjugated anti-CD49f, FITC-conjugated anti-CD227, PerCP-Cy5-conjugated anti-CD326 (EpCAM), and PE-conjugated anti-CD309 (all Becton Dickinson, San Jose, CA, USA). The following isotype control groups were used: PerCPCy5.5 IgG1, APC IgG2b, APC-H7 IgG2b, PE IgG2a, FITC IgG1, and PE-Cy7 IgG1. All antibodies were titrated to determine their optimal staining concentration and appropriate isotype controls were used. Labeled cells were washed thoroughly with 500 μL of FACSFlow (Becton Dickenson, Franklin Lakes, NJ, USA, Cat# 342003).

All samples were run on a Becton Dickenson FACSCanto II flow cytometer. The instrument was set up and standardized using BD Cytometer Setup and Tracking (CS&T) procedures according to manufacturer specifications. Data were analyzed using FACSDiva 8.0 software.

### 2.6. Isolation of Breast Cancer Cells

The CD326^+^ and CD227^+^ cell fractions were isolated using the EasySep™ Human CD326 (StemCell Technologies, Canada, Vancouver, Catalog #18356) or CD227 (StemCell Technologies, Canada, Vancouver, Catalog #18359) positive selection kit according to the technical protocol supplied by StemCell Technologies (StemCell Technologies, Canada, Vancouver). For EasySep™ cell separation, the labeled cell suspension was placed in the EasySep™ magnet for 5 min, and the cells that were not magnetically labeled were discarded. Labeled cells were resuspended, and the separation was repeated a total of 6 times. The EasySep™ cell separation was evaluated by flow cytometry.

### 2.7. In Vitro Tumor Study

Figure 2 shows the experimental design of the in vitro investigation.

Separation-sorted human breast cancer cell concentrations were determined using a cell counter and then seeded at densities of 5 × 10^5^ cells/mL in low adherence 6-well plates. The cultures were maintained in Human EpiCult-C (StemCell Technologies, Vancouver, Canada), supplemented with 5% fetal bovine serum (FBS, Sigma-Aldrich, St. Louis, MO, USA) for 24 h, and then the medium was replaced with serum-free medium and maintained for an additional 10 days. At the end of the assays, the colonies were counted under a microscope. After 10 days, the medium was removed and the plates gently rinsed with PBS. The cultured cells were then counted. The procedure was repeated three times. Sorted cells were seeded and cultivated in MammoCult^TM^ (StemCell Technologies, Vancouver, Canada, Cat#05620) supplemented with 0.48 μg/mL freshly dissolved hydrocortisone (StemCell Technologies, Vancouver, Canada, Cat#07904) and 4 μg/mL heparin (StemCell Technologies, Vancouver, Canada, Cat#07980) and 10 ng/mL IL-6 (Sigma-Aldrich, St. Louis, MO, USA, Cat# SRP3096) to induce greater numbers of mammospheres and tumorspheres, and the cultures were maintained for an additional 7 to 10 days. At the end of the assay, cells were assessed by flow cytometry and image processing of each well with Cytation™ 3.

CD227^+^- and CD326^+^-sorted cell populations were cultivated in the presence of 10 ng/mL cytostatics (docetaxel (Taxotere (T)) + adriamycin (A) + cyclophosphamide (C), TAC). Levels of apoptosis were evaluated after 2 h culture using flow cytometry and image processing of each well with Cytation™ 3.

In the next stage, we chose a patient with a high grade of malignancy and a high level of ALDH1^+^ cells circulating in the blood. The patient received cycles of TAC-based chemotherapy. We studied the dynamics of the count of tumor cells in the blood before and after surgery, after courses of chemotherapy. Moreover, we studied in vitro culture tumor cells isolated from breast tumor tissue of this patient with and without cytostatics. For the in vitro study, cytostatics which the patient received were used. Figure 3 shows the experimental design of these investigations in vitro.

### 2.8. Imaging

Images of CD227^+^ and CD326^+^ cells were obtained using a Cytation 3 cell imaging multimode reader (BioTek Instruments, Inc., Winooski, VT, USA) tuned to DAPI, GFP, and Texas Red light cubes.

At the end of the incubation period, CD227^+^ and CD326^+^ were counter stained with Hoechst 33342 (blue) for cell number measurements, or Annexin V-iFluor™ 350 Conjugate and 7-AAD fluorescent probes for apoptosis/ necrosis assessments. Then, a Cytation 3 (magnification of 4× or 20×) was imaged, followed by cell analysis using Gen5™ data analysis software (Bad Friedrichshall, Germany).

All collected images were pre-processed to align the background before applying analytical methods. Cell analysis was performed on a blue channel to determine cell count based on the number of Hoechst-stained nuclei. The default settings resulted in adequately calculated data for further analysis.

### 2.9. Statistical Analysis

All statistical analyses were carried out by using SPSS statistical software (version 15.0, SPSS Inc., Chicago, IL, USA). Statistical analysis was performed using the Mann–Whitney U test. Additionally, *p* values < 0.05 were considered to indicate statistically significant differences. All quantitative data presented are the mean value and standard error.

## 3. Results and Discussion

The ideal way to identify and track disease recurrence and/or progression in cancer patients is through surrogate marker approaches that are minimally invasive, reliable, and allow for longitudinal testing of accessible sample types, such as blood. At present, no single molecular or cellular blood marker has been proven to have such properties in BC. The most commonly used molecular markers are carcinoembryonic antigen or products of the MUC-1 gene expressed by tumor cells [19]. The high heterogeneity of the CSC phenotype in cancer was the reason for the study of various oncoantigens in our study. When studying blood samples from patients with BC, we found an increase in the number of tumor cells with overexpression of CD227 and CD326, as well as CSCs of mesenchymal origin (CD44^+^CD24^−^) compared to the levels in healthy volunteers (Figure 4). Moreover, we observed an increase in the HSCs and CD309^+^ endothelial cells in the blood relative to healthy volunteers. In addition, in BC an increase in the number of ALDH1^+^ cells circulating in the blood was observed.

Our results are consistent with the literature, which indicates a correlation between the tumor progression and the number of HSCs and endothelial cells in the blood [15]. Therefore, it is important to investigate the potential presence of HSC and endothelial cells in the peripheral blood of patients with BC, in addition to the already known tumor markers.

The tumor fraction ALDH1 is widely used as a biomarker of metastatic activity and a determinant of the clinical outcome of breast cancer [20]. The collection of material for biopsy is a rather painful intervention. Notably, prognostic potential of ALDH1^+^ CTC in patients with BC was suggested in the literature [20,21,22]. ALDH expression by circulating tumor cells correlates with poor clinical outcome, metastatic progression, and the response to therapy in patients with metastatic breast cancer [22]. Another study suggested that ALDH1 expression in primary breast tumors correlates with the presence of CTCs and clinical outcome in patients with non-metastatic disease [22]. Objectively, ALDH1^+^ cells circulating in the blood can be used as diagnostic markers. In our study, we found that patients with BC differed in the content of ALDH1^+^ cells in the blood. We identified a group of patients with a significant number of ALDH1^+^ cells (≥0.9% of all isolated mononuclear cells, group ALDH1^hi^ cells) and a group of patients with a small number of ALDH1^+^ cells (≤0.9% of all isolated mononuclear cells, group ALDH1^low^ cells) (Figure 5). It is important to note that the level of ALDH1^+^ cells in the tumor of patients of ALDH1^hi^ cells group was also higher than that of patients of the ALDH1^low^ cells group.

In BC, metastases in their development go through the mesenchymal and epithelial phases of the metastatic cascade [22]. ALDH1 is involved in the epithelial and more proliferative phase of metastatic tissue colonization. In this regard, the use of ALDH1 as a key biomarker of the metastases risk may be more useful in the diagnosis of advanced breast cancer (epithelial phase) [22]. In patients with an unexpanded disease, which is characterized by an intravascular and more mesenchymal phase of the metastatic cascade, the level of ALDH1 expression is significantly reduced. We found confirmation of this in the present study. According to our data, a patient of the ALDH1^hi^ cells group had metastases in the lungs.

When determining other antigens, we found intergroup differences in other CSCs. The ALDH1^hi^ cells group showed an increased number of epithelial tumor cells CD44^+^CD24^low^, CD326^+^CD44^+^CD24^−^, and CD326^–^CD49f^+^ (Figure 6). At the same time, in the ALDH1^low^ cells group, the content of CSCs of mesenchymal origin (CD44^+^CD24^−^), epithelial tumor cells (CD227^+^CD44^+^CD24^−^ and CD44^+^CD24^−^CD49f^+^), HSCs, and CD309+ endothelial cells in the tumor prevailed.

Despite the increased level of ALDH1 in the patients with BC, we did not find an increase in the numbers of HSC and CD309^+^ endothelial cell levels in the blood or tumor tissue compared to patients with ALDH1^low^. It is known that endothelial cells play a role in tumor progression by promoting the progression of avascular micrometastasis to vascularized macrometastases [23]. HSCs play an important role in the development of inflammation and tumor progression [14].

We suggest that patients with different levels of ALDH1 in the blood can suffer from tumors at different stages. The populations of CSC, HSCs, and endothelial cells isolated in this study can be used as markers for personalized tumor therapy, and disease prognosis.

Since we did observe group differences in the content of tumor CD326^+^CD44^+^CD24^−^ cells and CD227^+^CD44^+^CD24^−^ cells from the tumor of patient A of the ALDH1^hi^ cells group and patient B of the ALDH^low^ cells group, we isolated CD326^+^ and CD227^+^ cells, obtained primary cultures, and characterized them. At the same time, we evaluated the effects of IL-6 (a factor that stimulates CSC [24]) and cytostatics proposed for chemotherapy (CT) in vitro.

CD227^+^ cells of patient B and CD326^+^ cells of patient A were investigated (Figure 7a). We found that in patient B, the culture of CD227^+^cells was characterized by the absence of cells in apoptosis, insignificant clonal activity (mammosphere) and an increase in cell mass during the cultivation cycle (Figure 7d–f). In culture of CD227^+^ cells of patient B, IL-6 increased the frequency of mammosphere formation. After co-cultivation with cytostatics, the proportion of apoptotic CD227^+^ cells was 70% (Figure 7b,c).

The culture of CD326^+^ cells of patient A withstood three passages and by the end of the cultivation cycle its cell mass significantly increased. Moreover, the frequency of mammosphere formation increased in the presence of IL-6 (Figure 7a). The activity of the cells of patient A in culture was superior to those of patient B. On the other hand, patient A’s CD326^+^ cells were resistant to cytostatics: the number of cells in apoptosis was 14% versus 27% in patient B.

These data made it possible to suggest that when diagnosing breast cancer, it is desirable to determine the level of ALDH1^+^ cells circulating in the blood and the ratio of CD326^+^ and CD227^+^ tumor cells in the tumor. These cells can act as tumor progression factors and inducers of metastases in all groups, regardless of ALDH1 expression. To increase the effectiveness of chemotherapy, we suggest in vitro selection of cytostatics. This approach can enhance effective patient care.

In the next stage, we studied the count of tumor cells in the blood before and after surgery, after courses of chemotherapy of the patient received cycles of TAC-based chemotherapy (Figure 8a). We observed a decrease in the content of CD227^+^CD44+CD24^−^, CD227^+^, and CD44^+^CD24^low^ cells in the blood after surgery. However, we observed an increase in the content of these cell populations after courses of chemotherapy. We suggest that changes in blood can be associated with the resistance of cancer cells to chemotherapy. We studied in vitro cultured CSCs and tumor cells isolated from breast tumor tissue of this patient with and without cytostatics (Figure 8b–d). We found no changes in the count of dead tumor cells and apoptotic tumor cells. Moreover, no differences were observed in the count of tumor cells (CD326^−^CD49^+^, CD326^+^CD227^+^, CD326^+^CD44^+^CD24^−^, CD227^+^, CD227^+^CD44^+^CD24^−^) after a cycle of cultivation with cytostatics (TAC) treatment.

Thus, dynamic control of circulating CSCs of patients with BC makes it possible to detect cells with the potential for tumor progression and metastasis in the blood. In addition, in vitro assessment of the sensitivity of the CSC to the chemotherapy may allow a quick correction of the patient’s treatment.

In conclusion, it should be noted that in the breast tissue adjacent to the tumor (normal tissue) and tumor tissue of patients with BC, different populations of epithelial progenitor cells of a healthy mammary gland were revealed (Figure 9). We did not find significant differences in their content between the patients of the ALDH1^hi^ and ALDH1^low^ groups. In all groups, cells were associated with a positive effect of chemotherapy and remission of patients with BC. Thus, targeting epithelium and endothelium regeneration of the mammary gland might be beneficial and prevent tumors. However, our study has some limitations. Since we performed a short-term single-center study, it has a relatively small sample size, which diminishes the likelihood of generalization. To assess their applicability to a larger population, the results presented here need further validation using multi-center cohorts with large numbers of patients.

## 4. Conclusions

Patients with IIA–IIIB (T1-4N0-3M0), triple negative BC, and HER+ BC, are divided into a group with a significant number of ALDH1^+^ cells and a group with a small number of ALDH1^+^ cells in the blood and tumors. The composition of tumor CSCs and their activity in patients of the ALDH1^hi^ cells group and the ALDH1^low^ cell group differ. ALDH1 expression level and ratio of tumor cells CD44^+^CD24^low^, CD326^+^CD44^+^CD24^−^, CD326^−^CD49f^+^, CSC of mesenchymal origin (CD44^+^CD24^−^) and epithelial tumor cells (CD227^+^CD44^+^CD24^−^ and CD44^+^CD24^−^CD49f^+^), HSC, and CD309^+^ endothelial cells in tumors can act as personalized diagnostic markers, predictors of complications and the effectiveness of breast cancer treatment in further research. Moreover, the dynamic control in the blood and assessment of the sensitivity of CSCs to cytostatics in vitro can improve the effectiveness of chemotherapy in BC.

## Figures and Tables

**Figure 1 biomedicines-09-01223-f001:**
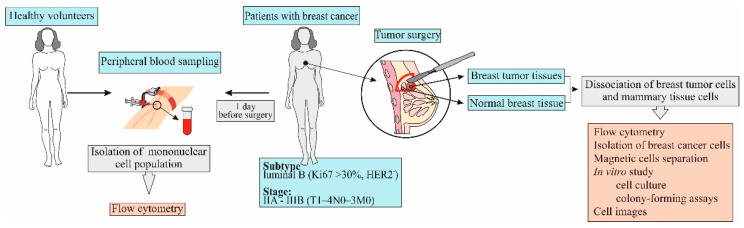
The experimental design of investigation of patients and volunteers.

**Figure 2 biomedicines-09-01223-f002:**
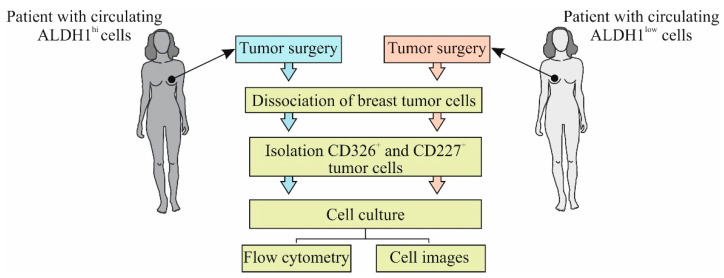
The experimental design of investigation in vitro.

**Figure 3 biomedicines-09-01223-f003:**
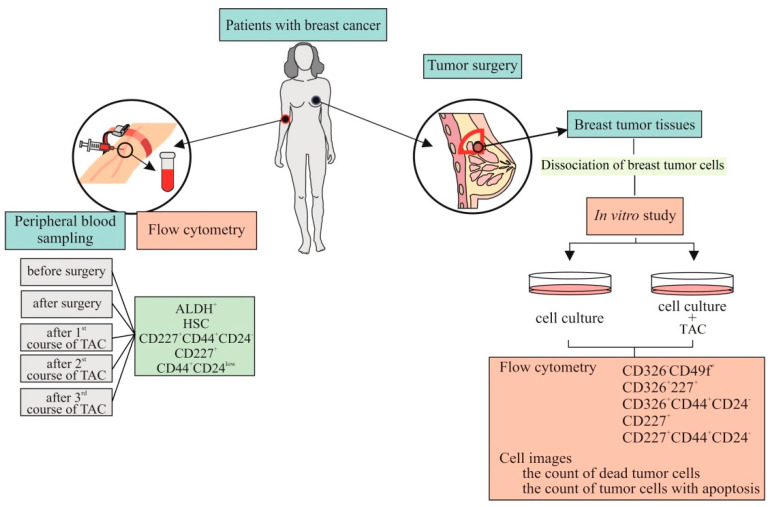
The experimental design of investigation patient with BC (blood and culture of cells isolated from breast tumor tissue). TAC is combination of cytostatics (docetaxel (taxotere (T)), adriamycin (A) and cyclophosphamide (C)).

**Figure 4 biomedicines-09-01223-f004:**
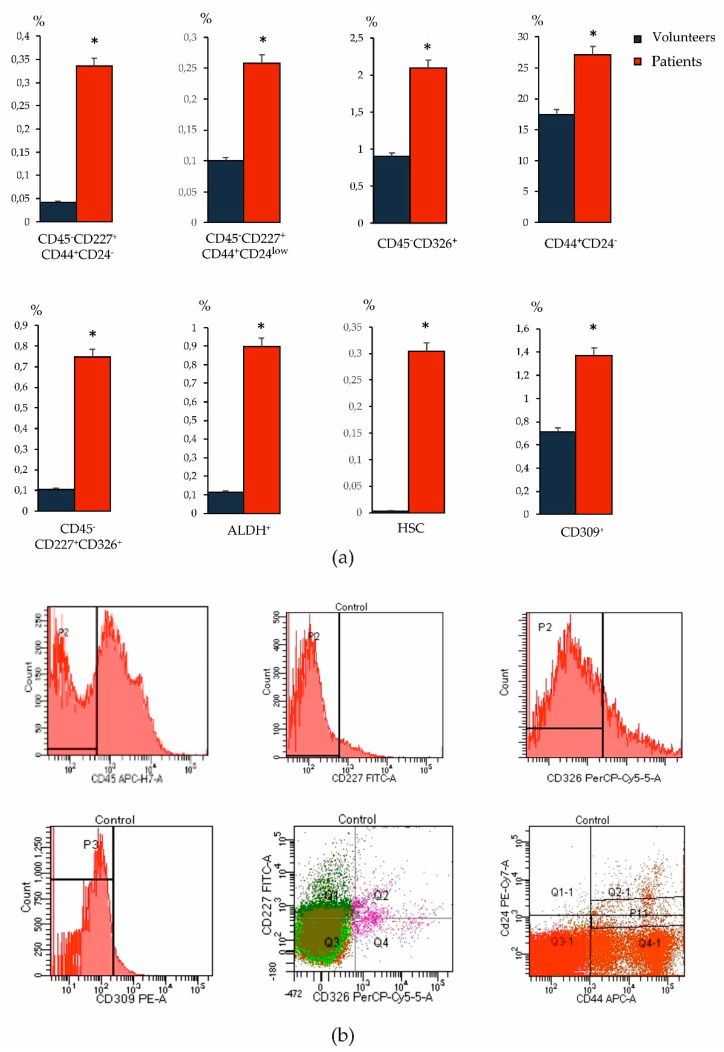
Characterization of epithelial tumor cells and tumor stem cells population, HSC (CD45^+^CD34^+^) and CD309^+^ endothelial cells isolated from blood of patients with BC and healthy volunteers. Cells were analyzed by flow cytometry using antibodies for CD24, CD34, CD44, CD45, CD227, CD309, CD326, and ALDH1. We used the two-color research reagent CD45/CD34 to determine HSC. We determined HSC according to ISHAGE protocol. (**a**) The level of epithelial tumor cells (CD45^−^CD227^+^CD44^+^CD24^−/low^), tumor stem cells (CD44^+^CD24^−^), HSC (CD45^+^CD34^+^) and CD309^+^ endothelial cells in the blood of patients with BC and healthy volunteers; (**b**) phenotype establishment and qualitative analysis of CD45 (APC-H7), CD227 (FITC), CD326 (PerCP-Cy5.5), CD24 (PE-Cy7), CD44 (APC). * Differences are significant in comparison with healthy volunteers (*p* < 0.05).

**Figure 5 biomedicines-09-01223-f005:**
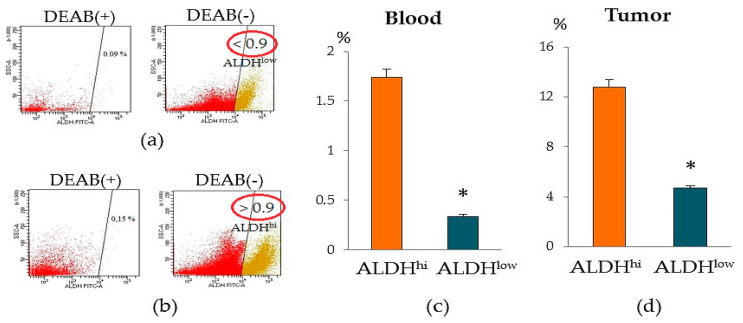
Isolation of circulating ALDH1^+^ cells from the blood of BC patients. (**a**) The patients with BC were divided into two distinct groups based on the level of aldehyde dehydrogenase positive cells (ALDH1^+^ cells) in the blood. (**a**) Aldefluor FACS analysis of the BC patients with ≤0.9 % of all isolated mononuclear cells (ALDH1^low^); (**b**) Aldefluor FACS analysis of the BC patients with ≥0.9 % of all isolated mononuclear cells (ALDH1^hi^); (**c**) the level of circulating ALDH1^+^ cells; (**d**) the level of ALDH1^+^ cells in tumor. * Differences are significant in comparison with the ALDH1^hi^ group (*p* < 0.05).

**Figure 6 biomedicines-09-01223-f006:**
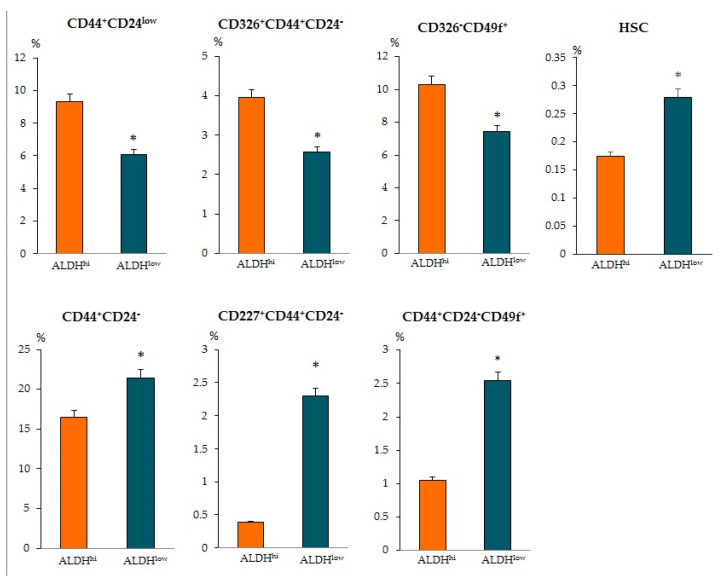
Characterization of HSC, epithelial tumor cells and tumor stem cells population isolated from tumor of patients with BC. Cells were analyzed by flow cytometry using antibodies for CD24, CD44, CD49f, CD227, CD326, and CD45/CD34 (for HSC). The patients with BC were divided into two distinct groups based on the level of aldehyde dehydrogenase positive cells (ALDH1^+^ cells) in the blood (≥0.9% of all isolated mononuclear cells, ALDH1^hi^ and ≤0.9% of all isolated mononuclear cells, ALDH1^low^). * Differences are significant in comparison with the ALDH1^hi^ group (*p* < 0.05).

**Figure 7 biomedicines-09-01223-f007:**
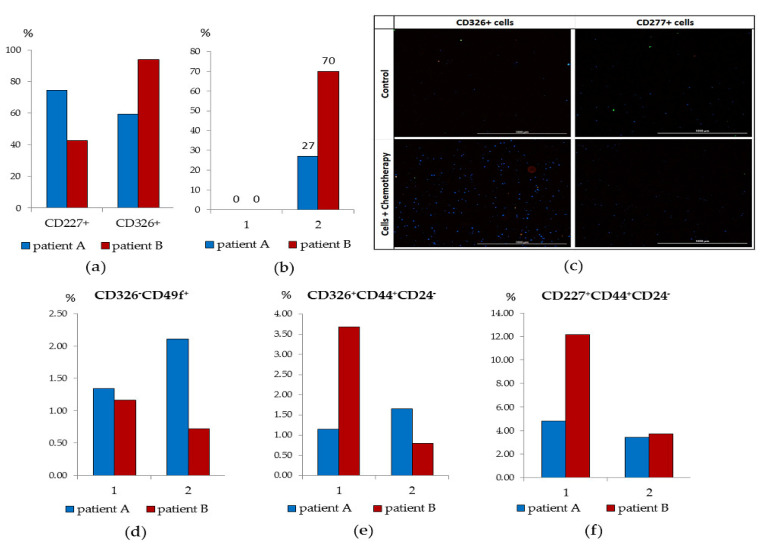
In vitro tumor cell activity from patients A and B. (**a**) The content of tumor stem cells in the CD227^+^- and CD326^+^-cell enriched environment after a cycle of cultivation and IL-6 treatment (% of the initial cell content in each population); (**b**) the count of tumor cells with apoptosis after a cycle of cultivation without cytostatic (1) and with cytostatics (docetaxel (taxotere (T)), adriamycin (A) and cyclophosphamide (C)), (2) treatment in the CD326^+^ cell population of patient A and CD227^+^ cell population of patient B; (**c**) 20× images of CD227^+^ and CD326^+^ cells stained with: Hoechst (blue) to identify cell nuclei; annexin V (green); 7-AAD (red); (Hoechst + Annexin V + 7-AAD) composite image using all three colors. Determination of the percentage of cells in apoptosis was made by the ratio of cells counted in green and red channel to total cells counted using blue (DAPI) channel. All scale bars are 1000 µm; (**d**) the count of tumor cells (CD326^−^CD49^+^) after a cycle of cultivation without cytostatic (1) and with cytostatics (docetaxel (taxotere (T)), adriamycin (A) and cyclophosphamide (C)) (2) treatment in the CD326^+^ cell population of patient A and CD227^+^ cell population of patient B; (**e**) The count of tumor cells (CD326^+^CD44+CD24^−^) after a cycle of cultivation without cytostatic (1) and with cytostatics (docetaxel (taxotere (T)), adriamycin (A) and cyclophosphamide (C) (2) treatment in the CD326^+^ cell population of patient A and CD227^+^ cell population of patient B; (**f**) the count of tumor cells (CD227^+^CD44^+^CD24^−^) after a cycle of cultivation without cytostatic (1) and with cytostatics (docetaxel (taxotere (T)), adriamycin (A) and cyclophosphamide (C)) (2) treatment in the CD326^+^ cell population of patient A and CD227^+^ cell population of patient B.

**Figure 8 biomedicines-09-01223-f008:**
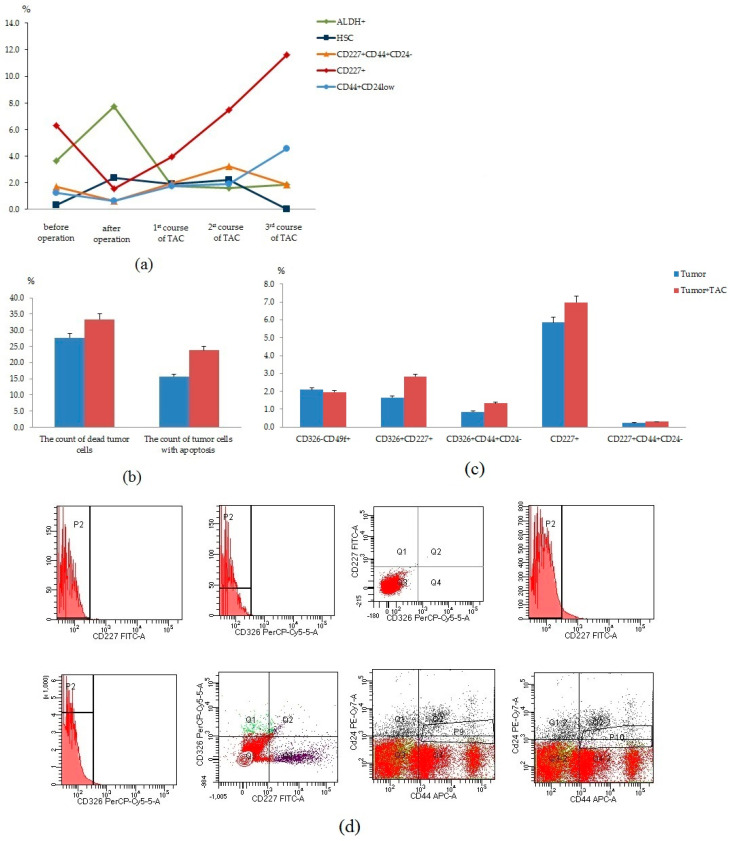
Characterization of tumor cells isolated from blood and tumor of patient with BC. (**a**) The content of ALDH1^+^, HSC, CD227^+^CD44^+^CD24^−^, CD227^+^, and CD44^+^CD24^low^ cells in the blood before and after surgery, after courses of chemotherapy. Cells were analyzed by flow cytometry using antibodies for human CD24, CD44, CD227, CD45, CD34, and ALDH1; (**b**) the count of dead tumor cells and tumor cells with apoptosis after a cycle of cultivation without cytostatic (tumor) and with cytostatics (docetaxel (taxotere (T)), adriamycin (A) and cyclophosphamide (C)) (tumor + TAC) treatment. Determination of the percentage of dead cells and cells in apoptosis was made by the ratio of cells counted in green and red channel to total cells counted using blue (DAPI) channel; (**c**) the count of tumor cells (CD326^−^CD49^+^, CD326^+^CD227^+^, CD326^+^CD44^+^CD24^−^, CD227^+^, CD227^+^CD44^+^CD24^−^) after a cycle of cultivation without cytostatic (tumor) and with cytostatics (docetaxel (Taxotere (T)), adriamycin (A) and cyclophosphamide (C)) (tumor + TAC). Cells were analyzed by flow cytometry using antibodies for human CD24, CD44, CD227, and CD326; (**d**) Isotype controls and phenotype establishment and qualitative analysis of CD227 (FITC), CD326 (PerCP-Cy5.5), CD24 (PE-Cy7), CD44 (APC) expression.

**Figure 9 biomedicines-09-01223-f009:**
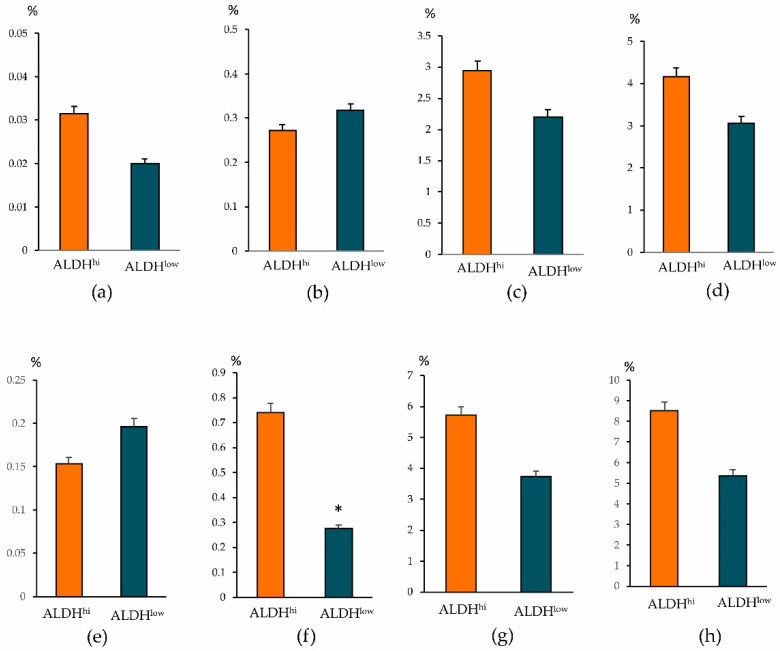
Characterization of bipotent precursors of breast cells (CD326^low^CD49f^hi^CD227^+^) (**a**,**e**), basal tissue-specific stem cells of the mammary gland (CD326^hi^CD49f^hi^) (**b**,**f**), precursors of luminal cells (CD326^low^CD49f^low^CD227^+^) (**c**,**g**), and precursors of myoepithelial cells (CD326^low^CD49f^+^) (**d**,**h**) isolated from the breast tissue adjacent to the tumor (normal tissue) (**a**–**d**) and tumor tissue (**e**–**h**) of patients with BC of the ALDH1^hi^ and ALDH1^low^ groups. Cells were analyzed by flow cytometry using antibodies for CD49f, CD227, and CD326. * Differences are significant in comparison with the ALDH1^hi^ group (*p* < 0.05).

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
