# Peer review of "Cancer Stem Cells and Somatic Stem Cells as Potential New Drug Targets, Prognosis Markers, and Therapy Efficacy Predictors in Breast Cancer Treatment"

_biomedicines, 2021, doi:10.3390/biomedicines9091223_

Round 1

Reviewer 1 Report

1. It is recommended to provide clinical data and prognosis of breast cancer patients in the study (2017-2018) to check whether they are consistent with this study. The article only mentions one metastasis.

2. Although trying to quote previous studies to express that the results of the study are related to the prognosis of treatment, the main experiments still use traditional chemical drugs. Whether it can be applied to new drugs and prognostic evaluation is still controversial. Need to modify the title of the article.

Author Response

We thank the reviewer for their time and valuable comments. We have now revised the manuscript according to the suggestions. Please find below the reviewers’ comments and our responses. All changes have been included in the revised manuscript.

Comments and Suggestions for Authors

  1. It is recommended to provide clinical data and prognosis of breast cancer patients in the study (2017-2018) to check whether they are consistent with this study. The article only mentions one metastasis.

- We now added information about patients in Supplementary Table 1. The article mentions metastasis was found because we evaluated dynamically CSC in blood after treatment.

  1. Although trying to quote previous studies to express that the results of the study are related to the prognosis of treatment, the main experiments still use traditional chemical drugs. Whether it can be applied to new drugs and prognostic evaluation is still controversial. Need to modify the title of the article.

- Our study was pilot.  All patients received standard treatment. We propose that our approach can be applied to new drugs and prognostic evaluation. We know that our study has limitations. We now plan to continue our study to confirm our approach. We modified the title of the article. 

Reviewer 2 Report

In the manuscript entitled "Cancer Stem Cells and Somatic Stem Cells as Potential New Drug Targets, and Prognosis Markers, and Therapy Efficacy Predictors in Breast Cancer Treatment" authors describe the results of a pilot study where they search for BC biomarkers which could improve BC therapeutic handling.

Authors are aware of their study limitations as sample size is really small jeopardizing generalization of their findings. With only 12 patients evaluated, which presented both luminal B and TNBC and were spread in wide age range (38 to 66 years) study misses a correlation between possible BC subtypes or age related outcome.

Major point:

- Authors must provide in text or as supplementary material, a table with all individual measurements for each patient so that reader can have an overview of all parameters for a single patient relating also to BC subtype and age.

Minor points:

- Please make sure that the resolution of figures is appropriate for publication.

- On item 2.7, could authors explain why cells were kept in starvation (serum free) during the experimental 10 days?

Author Response

We thank the reviewer for their time and valuable comments. We have now revised the manuscript according to the suggestions. Please find below the reviewers’ comments and our responses. All changes have been included in the revised manuscript.

We understand that our study has limitations as the sample size is really small jeopardizing. Therefore maybe we did not observe a correlation between possible BC subtypes or age-related outcomes. We now plan to continue our studies.

Major point:

- Authors must provide in text or as supplementary material, a table with all individual measurements for each patient so that reader can have an overview of all parameters for a single patient relating also to BC subtype and age.

We now have corrected it. We now added information about patients in Supplementary Table 1.

Minor points:

- Please make sure that the resolution of figures is appropriate for publication.

We now have corrected figures.

- On item 2.7, could authors explain why cells were kept in starvation (serum free) during the experimental 10 days?

Thank you for the interesting question. We used the technical protocol supplied by StemCell Technologies (EpiCult™-B or EpiCult™-C Medium (Human), StemCell Technologies, Vancouver, Canada) for culture. The cells were kept in starvation (serum-free) during the experimental 10 days according to the technical protocol supplied by StemCell Technologies.  It was known that change to serum medium could result in overgrowth of the culture by contaminating stromal cells.

Round 2

Reviewer 1 Report

Accept in present form

Reviewer 2 Report

Authors have provided the requested revisions and supplementary material. Manuscript is now considered appropriate for publication at Biomedicines.